# Plasma Phospholipid Fatty Acids and Coronary Heart Disease Risk: A Matched Case-Control Study within the Women’s Health Initiative Observational Study

**DOI:** 10.3390/nu11071672

**Published:** 2019-07-21

**Authors:** Qing Liu, Nirupa R. Matthan, JoAnn E. Manson, Barbara V. Howard, Lesley F. Tinker, Marian L. Neuhouser, Linda V. Van Horn, Jacques E. Rossouw, Matthew A. Allison, Lisa W. Martin, Wenjun Li, Linda G. Snetselaar, Lu Wang, Alice H. Lichtenstein, Charles B. Eaton

**Affiliations:** 1Department of Epidemiology, School of Public Health, Brown University, Providence, RI 02912, USA; 2Jean Mayer USDA Human Nutrition Research Center on Aging, Tufts University, Boston, MA 02111, USA; 3Brigham and Women’s Hospital, Harvard Medical School, Boston, MA 02115, USA; 4MedStar Health Research Institute, Hyattsville, MD 20782, USA; 5Georgetown-Howard Universities Center for Clinical and Translational Science, Washington, DC 20057, USA; 6Division of Public Health Sciences, Fred Hutchinson Cancer Research Center, Seattle, WA 98109, USA; 7Department of Preventive Medicine, Fineberg School of Medicine, Northwest University, 680 N Lake Shore Drive, Suite 1400, Chicago, IL 60611, USA; 8National Heart, Lung, and Blood Institute, Bethesda, MD 20892, USA; 9University of California, La Jolla, San Diego, CA 92093, USA; 10VA San Diego Healthcare System, San Diego, CA 92161, USA; 11Division of Cardiology, George Washington University School of Medicine and Health Sciences, Washington, DC 20052, USA; 12Health Statistics and Geography Lab, Division of Preventive and Behavioral Medicine, University of Massachusetts Medical School, 55 Lake Avenue North, Worcester, MA 01655, USA; 13Department of Epidemiology, University of Iowa, 200 Hawkins Drive, Iowa City, IA 52242, USA; 14Center for Primary Care and Prevention, Kent Hospital, Pawtucket, RI 02860, USA; 15Department of Family Medicine, Brown University Alpert Medical School, Providence, RI 02903, USA

**Keywords:** plasma phospholipid fatty acids, coronary heart disease, postmenopausal women

## Abstract

Background and Aims: The association of fatty acids with coronary heart disease (CHD) has been examined, mainly through dietary measurements, and has generated inconsistent results due to measurement error. Large observational studies and randomized controlled trials have shown that plasma phospholipid fatty acids (PL-FA), especially those less likely to be endogenously synthesized, are good biomarkers of dietary fatty acids. Thus, PL-FA profiles may better predict CHD risk with less measurement error. Methods: We performed a matched case-control study of 2428 postmenopausal women nested in the Women’s Health Initiative Observational Study. Plasma PL-FA were measured using gas chromatography and expressed as molar percentage (moL %). Multivariable conditional logistic regression was used to calculate odds ratios (95% CIs) for CHD associated with 1 moL % change in PL-FA. Results: Higher plasma PL long-chain saturated fatty acids (SFA) were associated with increased CHD risk, while higher n-3 polyunsaturated fatty acids (PUFA) were associated with decreased risk. No significant associations were observed for very-long-chain SFA, monounsaturated fatty acids (MUFA), PUFA n-6 or *trans* fatty acids (TFA). Substituting 1 moL % PUFA n-6 or TFA with an equivalent proportion of PUFA n-3 were associated with lower CHD risk. Conclusions: Higher plasma PL long-chain SFA and lower PUFA n-3 were associated with increased CHD risk. A change in diet by limiting foods that are associated with plasma PL long-chain SFA and TFA while enhancing foods high in PUFA n-3 may be beneficial in CHD among postmenopausal women.

## 1. Introduction

The relationship between fat intake and heart disease is not entirely clear, particularly among postmenopausal women with relatively low fat consumption [1,2]. Some meta-analyses of observational studies and randomized controlled trials found no association between saturated fatty acids (SFA) and risk of coronary heart disease (CHD) [3,4]. However, these studies lack a consideration of the nutrient substitution framework, under which significant associations have been observed [5,6,7]. Dietary intervention trials have demonstrated that SFA elevate total cholesterol and low-density lipoprotein cholesterol (LDL-C), and substituting SFA with polyunsaturated fatty acids (PUFA) is associated with lower CHD risk [8,9]. Based on the above evidence, several professional organizations have recommended diets limiting SFA and increasing PUFA n-3 [10,11].

Until recently, all SFA were viewed as unhealthy as refined carbohydrates when considering their impact on increasing LDL-C and total cholesterol. Although not all SFA have the same cholesterol-raising effect, no recommendations have been made for specific SFA due to insufficient evidence. Evidence supporting the associations of monounsaturated fatty acids (MUFA) and PUFA n-6 with CHD risk is also less definitive [4,12]. A recent meta-analysis of prospective observational studies and randomized trials concluded that there was no significant association of dietary MUFA and PUFA n-6 with CHD risk, but a higher CHD risk of *trans* fatty acids (TFA) (pooled RR (95% Cis): 1.16 (1.06–1.27)) and a lower CHD risk of long-chain PUFA n-3 (0.87 (0.78–0.97)) [4]. Several investigators have questioned the interpretation of these results [9,13]. One major concern was that the macronutrient for replacement, which could have had an independent association with CHD outcome, was not identified or accounted for [9,13]. An additional concern was that the dietary assessment methods in these reports were via self-report instruments, which is valid when ranking individuals according to the intake levels, but contributes substantial systematic measurement error when the absolute intake levels are of interest.

To address these concerns, the Multi-Ethnic Study of Atherosclerosis [12,14] and the Women’s Health Initiative Observational Study (WHI-OS) [15] have measured plasma phospholipid fatty acids (PL-FA) as indicators of medium-term dietary FA intake [16,17], and have shown associations between PL-FA and CHD risk. Specifically, one interquartile range increase in PL long-chain PUFA n-3 or SFA 15:0 were associated with a lower CHD risk (HR (95%CIs), 0.40 (0.23–0.69) and 0.76 (0.61–0.93), respectively) [12,14]; 1 moL % increase in PL SFA was associated with a 20% higher CHD risk (95% CIs 1.08–1.32); and 1 moL % increase in PL PUFA n-3 was associated with a 11% lower CHD risk (95% CIs 0.83–0.97) [15]. While plasma PL-FA offer an objective biomarker of dietary FA that could potentially reduce the measurement error from questionnaires or diet recalls, plasma PL-FA reflect both dietary intake and endogenous synthesis. Therefore, caution is needed when interpreting the association with disease outcome.

Large observational studies and dietary intervention trials have demonstrated that PL-FA profiles are good biomarkers of dietary fat, especially for those FA with limited in vivo synthesis—PUFA n-3, n-6, and TFA [16,18]. In addition, randomized trials have shown that substituting dietary PUFA n-6 with PUFA n-3 at 10% of total energy led to a 2–3% plasma substitution [19]. Thus, the substitution of plasma PL-FA by changes in dietary fat types has a potential for evaluating the association of specific dietary FA with CHD risk, with less measurement error. However, the substitution of plasma PL-FA has not been examined so far, and we are still lack evidence from postmenopausal women.

Based upon the previously reported associations between plasma PL-FA and CHD, we sought to examine the following: (1) the association of plasma PL-FA levels, specifically long-chain and very-long-chain plasma PL SFA, with CHD risk; (2) the effect of substituting plasma PUFA n-6 and TFA with PUFA n-3 on CHD risk; and (3) potential food groups that are correlated with plasma PL-FA, using data from the WHI study.

## 2. Materials and Methods

### 2.1. Study Population

The WHI-OS is a prospective cohort study that enrolled 93,676 postmenopausal women between the ages of 50 and 79 years in the United States from 1994 to 1998. It is designed to assess the biological, lifestyle, and genetic factors for CHD and other major health events among postmenopausal women. A detailed description of the WHI study design has been published elsewhere [20,21].

A matched case-control design was used for the current study. All CHD cases, including hospitalized myocardial infarction (MI), definite silent MI, and coronary death, were confirmed based on medical records and death certificates. The cases included in the current study were a random sample of all CHD cases identified based on the September 2005 database [22]. A total of 2468 cases were initially selected and those who met the following criteria were excluded: (1) lack of sufficient baseline plasma sample (*N* = 28) (2) missing baseline dietary measurement (*N* = 126), and (3) self-reported baseline CVD, which includes angina, MI, coronary artery bypass graft (CABG) surgery, percutaneous transluminal coronary angioplasty (PTCA), carotid artery disease, congestive heart failure, stroke or peripheral vascular disease (*N* = 765). Potential controls were women from the entire WHI-OS who did not develop CVD during the follow-up (a mean of 4.5 years) and were excluded if meeting the same exclusion criteria as the cases. Cases and controls were matched on age at screening, enrollment date, race/ethnicity (White, Black, Hispanic, other), and hysterectomy status. The sample size for PL-FA assays included 1224 matched case-control pairs and 10% blind duplicates for quality control. We additionally excluded 20 participants due to the lack of plasma PL-FA profile results (*N* = 11) or missing matched pairs (*N* = 9), and came up with a final sample size of 1214 matched pairs. A separate approval for using de-identified samples and data for this study was obtained from the Tufts University/Tufts Medical Center Institutional Review Board [15].

### 2.2. Plasma PL-FA Profiles

Blood samples were collected at baseline and a minimum of 12 h fasting before blood draw was required. All blood samples were maintained at 4 °C for up to an hour until plasma was separated from cells, frozen at −20 °C, and then sent to the central repository stored at −80 °C. Plasma PL-FA profiles were measured by an established gas chromatography method [23] at Tufts University. Peaks of interest were identified by comparison with authentic FA standards (National Institute of Health Fatty Acid Standards A, B, and C, Nu-Check-Prep, Elysian, MN, USA), and expressed as molar percentage (moL %) proportions of FA relative to the internal standard (heptadecanoic acid). Internal and external quality controls were performed to guarantee the validity of the measurements, and detailed information has been published previously [15]. A total of 28 individual plasma PL-FA were measured. We classified these FA into groups based on the number of double bonds—specifically SFA, MUFA, PUFA n-3, PUFA n-6, and TFA. We further classified FA by the length of FA chains: long-chain FA are those with 12–19 carbons and very-long-chain FA are those with 20 or more carbons. Table 1 shows the lipid names, common names, categories, and mean (SD) levels of individual plasma PL-FA that were included in this study.

### 2.3. Covariates and Dietary Data

Standard questionnaires following the same protocol were utilized throughout the study to collect information related to socio-demographics, lifestyle factors, and CHD risk factors [20]. We initially considered the following variables as potential confounders: (1) socio-demographic variables (including age, U.S. region, race/ethnicity, education, and income); (2) lifestyle factors such as recreational physical activity, body mass index (BMI), waist circumference, waist-to-hip ratio, and smoking; and (3) CHD risk factors (family history of MI/diabetes/stroke, anticoagulant/anti-diabetic/lipid lowering medication use, postmenopausal hormone use, and self-reported hypertension/diabetes/hypercholesterolemia/hysterectomy status at baseline).

Information on nutrient intake and food consumption was assessed by food frequency questionnaire at baseline [20]. Potential dietary confounders included alcohol intake (g/day), percent calories from protein/carbohydrates, and total energy intake (kcal/day) [24]. Foods high in fats or that have the potential to influence fat metabolism were assessed for further correlation analysis with plasma PL-FA. For this analysis, we considered the following nine food groups: fish, dairy products, butter, margarine, olive/canola oil, other vegetable oils, red meat, alcohol, and carbohydrates [25,26,27,28].

Age and BMI were treated as continuous variables. Physical activity, measured by recreational physical activity score (MET-h/week) based on a series of questions related to exercise intensity levels [29], was treated as continuous or categorical dichotomized at median. Education was categorized as ≤high school, some college, or postgraduate. Income was categorized as <$20,000, $20,000–74,999, or ≥$75,000 per year. Smoking was categorized as current, past, or never smoker. Family history was defined as first-degree relatives having MI, diabetes, or stroke. Postmenopausal hormone use was categorized into current estrogen + progesterone, current estrogen alone, past users, or never used. Hypertension was defined as self-reported hypertension/taking antihypertensive medication, or systolic blood pressure ≥ 140 mm Hg and/or diastolic blood pressure ≥ 90 mm Hg. Baseline diabetes and hypercholesterolemia were defined as taking anti-diabetic or cholesterol-lowering medications, respectively.

### 2.4. Statistical Analysis

We initially examined the baseline distribution of socio-demographics, lifestyle factors, CHD risk factors, and dietary factors by CHD status, as well as by five subtypes of plasma PL-FA in tertiles (Appendix A). Descriptive statistics such as median, mean, standard deviation, frequency, and proportion were used to summarize the aforementioned variables. Depending on the distribution of the variables, we used the paired *t*-test, Wilcoxon signed rank test, or McNemar test for the comparison between cases and controls.

We employed multivariable conditional logistic regression models to estimate odds ratios (ORs) and corresponding 95% confidence intervals (CIs) for CHD risk in association with a 1 moL % increase in plasma PL-FA. The covariates in the multivariable model were selected based on a hypothesized causal diagram (Appendix B) to adjust for potential confounding, and a backward selection method was used to generate a parsimonious model with the best model fit [30]. The final multivariable model adjusted for matching factors (age, race/ethnicity, enrollment date, and hysterectomy status), income, physical activity, smoking, family history of MI/diabetes, postmenopausal hormone use, self-reported hypertension/diabetes, percent calories from protein/carbohydrates, and total energy intake. We have performed model diagnosis and examined model assumptions. No outlier was observed and all model assumptions held.

Linoleic, α-linolenic, and *trans* FA cannot be synthesized in vivo. In addition, the elongase and desaturase enzymes in human livers have low activity when regulating the synthesis of long-chain PUFA from their precursors [31]. Therefore, plasma PL PUFA n-3, n-6, and TFA are good biomarkers of corresponding dietary FA, and substitutions of dietary FA can be estimated by plasma substitutions. To estimate the theoretical effect of substituting 1 moL % of plasma PL PUFA n-6 with the same proportion of PUFA n-3, we left out PUFA n-6 in the multivariable model. The relation can be expressed as follows:*logit* (*CHD risk*) = *β*_1_ * *PL PUFA n*-3 + *β*_2_ * *PL SFA* + *β*_3_ * *PL MUFA* + *β*_4_ * *PL TFA*(1)
where $\beta_{1}$ to $\beta_{4}$ are regression coefficients. The total of all PL-FA is 100 moL %, so that the coefficient $\beta_{1}$ can be interpreted as the effect of substituting 1 moL % of PUFA n-6 with the same proportion of PUFA n-3 while holding other FA constant [24]. This substitution model was similarly applied to TFA substitution analysis.

To further examine potential food sources that related to plasma PL-FA levels, we calculated Spearman’s rank correlation coefficients between plasma PL-FA levels and the consumption of nine selected food groups. Considering the cases might have altered metabolic status, thus not representing the overall population, the correlation coefficients were calculated among controls only, adjusting for matching factors and the aforementioned non-dietary confounders.

Multiple imputation (five times) by chained equations [32] was used to impute missing values on the following covariates: income (*N* = 118), physical activity (*N* = 28), smoking (*N* = 28), family history (*N* = 209, among which 119 were missing MI and 122 were missing diabetes), and self-reported hypertension (*N* = 47) and diabetes (*N* = 2).

We conducted the following sensitivity analyses to assess the robustness of findings: (1) comparing the association analysis results from different regression models; (2) examining the association between plasma PL-FA groups and CHD additionally adjusting for anthropometric measures (BMI, waist circumference, waist-to-hip ratio) and chronic weight cycling (three-year BMI change); (3) performing substitution analysis stratified by physical activity levels; and (4) performing the association and substitution analyses among participants with complete information (*N* = 2181). All analyses were performed using Statistics Analysis Systems software package (version 9.4; SAS Institute, Inc., Cary, NC, USA).

## 3. Results

The characteristics of the cases and controls can be found in Table 2. The mean (SD) age was 67.8 (6.8) years, and the median time from baseline plasma PL-FA measures to CHD event among cases was 4.5 years. Compared with controls, cases had significantly lower education and income levels, higher BMI (26.9 vs. 25.9 kg/m^2^) and lower physical activity level (8.3 vs. 10.8 MET-h/week). More cases were smokers, had hypertension or diabetes, or reported a family history of MI and medication use, while fewer were currently using postmenopausal hormones.

### 3.1. Associations between Increased Plasma PL-FA and CHD Risk

To examine the relationships between plasma PL-FA and CHD risk, we calculated the ORs (95% CIs) of CHD in association with 1 moL % increase in plasma PL-FA (Table 3). In the adjusted multivariable model, we observed higher CHD risk for increased plasma total PL SFA (OR = 1.20 (1.10–1.30)) and long-chain SFA (OR = 1.18 (1.09–1.28)), but not for very-long-chain SFA (OR = 1.00 (0.77–1.30)). We also observed a lower CHD risk associated with plasma PL PUFA n-3 (OR = 0.93 (0.88–0.99)). However, no significant associations were observed for plasma PL MUFA, PUFA n-6, and TFA. The associations between individual plasma PL-FA and CHD have been published elsewhere [15].

### 3.2. Plasma PL-FA Substitutions

To further estimate the effect of substituting dietary PUFA n-6 or TFA with PUFA n-3 on CHD risk, we calculated the ORs (95%CIs) of CHD risk from plasma PL-FA substitutions (Table 4). In the initial models, lower CHD risk was observed when 1 moL % of plasma PL PUFA n-6 or TFA were substituted with the same proportion of PUFA n-3. Although these associations were attenuated on adjustment for covariates (model 2), results were still statistically significant at the 0.05 significance level (OR = 0.90 (0.84–0.96) and 0.74 (0.56–0.99), respectively). However, in all models, substituting 1 moL % PL TFA with PUFA n-6 was not associated with CHD risk.

### 3.3. Correlations between Plasma PL-FA and Select Food Groups

Correlations between food groups and plasma PL-FA levels have the potential to provide information regarding food sources that may influence plasma PL-FA concentrations (Appendix C). The strongest correlations that we found were in the PUFA n-3 group: fish and olive/canola oil intakes were positively correlated with plasma PL PUFA n-3 (*r* = 0.34 and 0.12, respectively; *p* < 0.0001). We also observed a positive correlation between alcohol intake and plasma PL long-chain SFA (*r* = 0.13; *p* < 0.0001). Margarine and red meat intakes were positively correlated with plasma PL PUFA n-6 and TFA (*r* ranges between 0.11 to 0.15; *p* < 0.0001).

### 3.4. Sensitivity Analyses

In the sensitivity analysis of comparing difference regression models, our results were consistent across models thus supporting the robustness of findings (Appendix D). When examining the association between plasma PL-FA and CHD while adjusting for different anthropometric measures, we found very similar results when adjusting for BMI, waist circumference, waist-to-hip ratio, or a three-year BMI change (Appendix E). In the substitution analysis stratified by physical activity levels, we classified participants into two groups: physically active (those with physical activity levels above the median, 9.5 MET-h/week) and physically inactive (those with physical activity levels ≤ 9.5 MET-h/week). As the point estimates varied to a small extent between the physically active and inactive, physical activity is less likely to be an effect modifier than a confounder (Appendix F). When comparing the association and substitution analyses using complete cases versus multiple imputation, we observed very similar results, thus suggesting the validity of the multiple imputation approach (Appendix G and Appendix H).

## 4. Discussion

This prospective matched case-control study nested in the WHI-OS assessed the association of plasma PL-FA profile with CHD risk among 2428 postmenopausal women. In the adjusted analysis, we found that higher PL SFA, especially long-chain SFA, were associated with increased CHD risk, while higher PL PUFA n-3 were associated with lower CHD risk. No significant associations were found for PL very-long-chain SFA, MUFA, PUFA n-6 and TFA. In the substitution analysis, we found that substituting 1 moL % of plasma PL PUFA n-6 or TFA with the same proportion of PUFA n-3 were associated with lower CHD risk.

### 4.1. Plasma PL SFA Profiles and CHD Risk

Individual SFA have diverse biological functions determined by the chain length [33]. For example, the effect of raising LDL-C decreases as the chain length increases [33]. Long-chain SFA, especially palmitic (16:0) and stearic (18:0) acids, are the primary dietary FA. Within the human body, long-chain SFA are a major component of cell membranes, and endogenous synthesis contributes a significant portion of SFA in the circulation with palmitic and stearic acids being the primary product [34]. Accumulating evidence has supported the relationship of the aforementioned long-chain FA metabolism with potential CHD risk [33]. The modest positive associations of total and long-chain PL SFA with CHD risk from our study were consistent with some [26,35], but not all [36,37], cohort studies that have assessed either total or individual SFA. The discordance may come from differences in age distributions, sources of blood SFA, and specific SFA included in analyses.

We did not observe significant associations between very-long-chain plasma PL SFA and CHD risk, which differs from a few recent population-based studies showing blood concentrations of very-long-chain SFA were associated with lower risk of cardiometabolic conditions, including CHD [38] and diabetes [39]. The mechanisms underlying these observations are not well established. Compared with long-chain SFA, very-long-chain SFA have lower water solubility and oxidation susceptibility, and they are major components of ceramides and sphingomyelins that affect liver homeostasis, myelin maintenance, and anti-inflammatory response through ceramide synthase expression, therefore showing potential beneficial effects on CHD [40,41]. The discrepancies between our results and other studies may be explained by the following: (1) differences in study populations: participants in other studies were younger and had lower BMI, and research has shown that the association of very-long-chain SFA with lower cardiometabolic risks appeared strongest in participants with normal BMI [39]; and (2) different sources of SFA: most beneficial effects were identified from circulating very-long-chain SFA, while no significant results were found from plasma PL or erythrocyte membranes.

### 4.2. Plasma PL-FA Substitution in Groups

We observed a lower risk of CHD when substituting plasma PL PUFA n-6 with n-3. The beneficial effects of PUFA n-3 on CHD have been reported in previous studies [12,14,15]. PUFA n-3 have been shown to lower plasma triglyceride levels, to have anti-thrombotic and anti-arrhythmic properties, to reduce macrophage infiltration into the vessel wall, and to reduce the proatherogenic secretion of growth factors and cytokines by monocytes [42].

Although dietary TFA have been shown to increase CHD risk [43], we only observed a significant change in CHD risk when substituting PL TFA with PUFA n-3, while no significant result was observed when TFA were substituted with PUFA n-6. This discrepancy may be explained by the limitation in the gas chromatography methodology we used to measure plasma PL-FA, which did not distinguish TFA isomers elaidic acid (18:1 n-9 t) and vaccenic acid (18:1 n-11 t) [15]. It has been suggested that 18:1 n-11 t, as the predominant TFA in dairy products, may have a weaker association with CHD compared with other TFA resulting from partial hydrogenated vegetable oils, mainly 18:1 n-9 t [44]. Another possibility is that the relatively small proportion of PL TFA, reflecting the dietary characteristics of the cohort of older women, lead to a lack of power to detect an association.

The substitution analysis between different plasma PL-FA may be more informative than the measured plasma PL-FA profiles themselves when examining the association of one type of plasma PL-FA with CHD. The plasma PL-FA were measured in moL % with a summation of 100% for all PL-FA, and 1 moL % increase in one type of plasma PL-FA is accompanied by reciprocal decrease in 1 moL % for another. The plasma PL-FA substitution analysis is also more informative than the substitution analysis of dietary FA and CHD risk, given that there is a lack of accuracy for each type of dietary fat using any diet assessment method [45].

### 4.3. Potential Food Sources of Plasma PL-FA and Dietary Recommendations

The Spearman correlations we observed between plasma PL-FA and selected food groups were moderate; however, they were consistent and comparable to previous evidence from other observational studies using food frequency questionnaire as dietary measurement [18,46]. In short-term dietary intervention trials and observational studies using multiple dietary measurements, the correlations were stronger, therefore showing the measurement error in food frequency questionnaire might be a cause of our moderate correlations [47,48].

The strongest correlation that we found was between plasma PL PUFA n-3 and fish intake. This result is consistent with both observational study and dietary intervention trials [18,46], showing fatty fish and fish oils are the predominant sources of PUFA n-3. We found a positive correlation between plasma PL long-chain SFA and alcohol intake. This may be because alcohol can increase the activity of acetyl-CoA carboxylase and FA synthase—the key enzymes in SFA 16:0 synthesis [49]. Our analysis also showed that margarine and red meat intake was positively associated with plasma PL TFA levels. Because humans do not synthesize TFA, diet contributes to the occurrence of these FA isomers in the plasma. TFA can be found naturally from ruminant-animal meat (mainly 18:1 n-7 t), dairy fat (mainly 18:1 n-11 t), and unnaturally from industrially hydrogenated vegetable oils (mainly 18:1 n-9 t), such as margarine [50].

Our findings are consistent with the recommendation made in the Scientific Report of the 2015 Dietary Guidelines Advisory Committee on food sources of nutrients among U.S. adults [51]. Foods associated with an increase of plasma PL long-chain SFA and TFA (including alcohol, red meat, and margarine) should be limited. Dietary PUFA n-3, which can be found in fatty fish and canola oil, should be recommended. However, caution is required when attempting to apply dietary recommendations extrapolated from plasma levels of nutrients, which are only modestly correlated with dietary intake.

There are several strengths of this study compared with previous ones. The matched case-control design nested in a prospective cohort study allowed us to assess exposure in cases before the diagnosis of CHD and select controls based on incidence density sampling, thus addressing the temporality between exposure and disease onset. In addition, our study is novel due to the focus on the association of theoretical plasma PL-FA substitutions on CHD risk, which reflect in vivo FA metabolism as well as dietary changes with less measurement error. However, our study had several limitations. We did not detect all plasma PL-FA (such as short- and medium-chain SFA with carbons less than 12 and isomers of TFA), thus limiting our analyses to a selective set of FA. However, these unmeasured FA are present in very small amounts. Additional limitations include: (1) confounding bias due to unmeasured CHD risk factors, such as blood lipids; and (2) limited evidence when extrapolating plasma PL-FA findings to dietary changes.

## 5. Conclusions

This plasma PL-FA analysis suggests that long-chain SFA may be associated with increased risk of CHD, and substituting PUFA n-6 or TFA with PUFA n-3 may be associated with lower risk of CHD. Further work is needed on distinguishing the specific dietary factors that influence plasma PL-FA levels within the context of the other covariates that may likewise impact outcomes.

## Figures and Tables

**Table 1 nutrients-11-01672-t001:** The lipid names, common names, categories, and mean (SD) levels (moL %) of plasma phospholipid fatty acid profiles measured in the matched case-control study of the Women’s Health Initiative (1994–2005) (*N* = 2428).

Lipid Names ^a^	Common Names	Category	Mean (SD) Levels (moL %)
SFA			46.09 (1.29)
12:0	Lauric acid	Long-chain	0.07 (0.04)
14:0	Myristic acid	Long-chain	0.69 (0.21)
15:0	Pentadecylic acid	Long-chain	0.23 (0.05)
16:0	Palmitic acid	Long-chain	30.57 (2.02)
18:0	Stearic acid	Long-chain	13.19 (1.42)
20:0	Arachidic acid	Very-long-chain	0.24 (0.07)
22:0	Behenic acid	Very-long-chain	0.64 (0.23)
24:0	Lignoceric acid	Very-long-chain	0.46 (0.18)
MUFA			11.77 (1.64)
14:1	5-myristoleic acid	Long-chain	0.12 (0.10)
16:1 n-9	7-palmitoleic acid	Long-chain	0.84 (0.32)
16:1 n-7	cis-9-palmitoleic acid	Long-chain	0.12 (0.05)
18:1 n-9	Oleic acid	Long chain	1.37 (0.28)
18:1 n-7	Vaccenic acid	Long-chain	8.48 (1.37)
20:1 n-9	cis-gondoic acid	Very-long-chain	0.08 (0.03)
24:1 n-9	Nervonic acid	Very-long-chain	0.77 (0.30)
PUFA			41.60 (2.06)
PUFA n-6			36.31 (2.46)
18:2 n-6	Linoleic acid (LA)	Long-chain	20.79 (3.00)
18:3 n-6	γ-linoleic acid	Long-chain	0.10 (0.05)
20:2 n-6	Eicosadienoic acid	Very-long-chain	0.40 (0.16)
20:3 n-6	Eicosatrienoic acid	Very-long-chain	3.32 (0.79)
20:4 n-6	Arachidonic acid (AA)	Very-long-chain	10.93 (2.07)
22:4 n-6	Docosatetraenoic acid	Very-long-chain	0.42 (0.11)
22:5 n-6	4,7,10,13,16-Docosapentaenoic acid	Very-long-chain	0.35 (0.13)
PUFA n-3			5.14 (1.58)
18:3 n-3	α-linolenic acid (ALA)	Long-chain	0.21 (0.08)
20:5 n-3	Eicosapentaenoic acid (EPA)	Very-long-chain	0.81 (0.51)
22:5 n-3	7,10,13,16,19-Docosapentaenoic acid (DPA)	Very-long-chain	0.83 (0.19)
22:6 n-3	Docosahexaenoic acid (DHA)	Very-long-chain	3.29 (1.13)
TFA			0.69 (0.34)
18:1t	All 18:1 *trans*	Long-chain	0.54 (0.30)
18:2t	All 18:2 *trans*	Long-chain	0.14 (0.06)

^a^ Lipid names are presented in the form of C:D, where C is the number of carbon atoms and D is the number of double bonds in the fatty acid. Abbreviations: MUFA (mono-unsaturated fatty acids), PUFA (polyunsaturated fatty acids), SD (standard deviation), SFA (saturated fatty acids), TFA (*trans* fatty acids).

**Table 2 nutrients-11-01672-t002:** Baseline characteristics by cases and controls in the Women’s Health Initiative study (1994–2005) (*N* = 2428).

Variables	Overall	CHD Status		
		Controls (*N* = 1214)	Cases (*N* = 1214)	*p*-Values ^a^
**Socio-demographics**				
Age, years ^b^	67.8 (6.8)	67.8 (6.8)	67.8 (6.8)	*Matched*
Race/ethnicity, *n* (%)				*Matched*
Black	136 (6)	68 (6)	68 (6)	
Hispanic	32 (1)	16 (1)	16 (1)	
White	2172 (90)	1086 (90)	1086 (90)	
Other	88 (4)	44 (4)	44 (4)	
Region, *n* (%)				*0.40*
Northeast	610 (25)	303 (25)	307 (25)	
South	579 (24)	299 (25)	280 (23)	
Midwest	547 (23)	264 (22)	283 (23)	
West	692 (29)	348 (29)	344 (28)	
Education, *n* (%)				*<0.01*
≤High school	845 (35)	380 (31)	465 (38)	
Some college and college graduate	907 (37)	471 (39)	435 (36)	
Postgraduate	676 (28)	363 (30)	314 (26)	
Income, *n* (%)				*<0.01*
<$20,000	448 (18)	193 (16)	255 (21)	
$20,000–$74,999	1541 (64)	776 (64)	765 (63)	
≥$75,000	439 (18)	245 (20)	194 (16)	
**Lifestyle factors**				
Physical activity, MET-h/week ^c^	9.5 (16.1)	10.8 (17.3)	8.3 (15.4)	*<0.01*
BMI, Kg/m^2^ ^c^	26.4 (6.8)	25.9 (6.4)	26.9 (7.4)	*<0.01*
Waist circumference, cm ^c^	84.2 (13.6)	83.0 (12.3)	86.5 (14.4)	*<0.01*
Waist-to-hip ratio ^c^	0.8 (0.1)	0.8 (0.1)	0.8 (0.1)	*<0.01*
Smoking, *n* (%)				*<0.01*
Never-smoker	1232 (51)	650 (54)	582 (48)	
Past smoker	1033 (43)	501 (41)	532 (44)	
Current smoker	163 (7)	63 (5)	100 (8)	
**CHD risk factors**				
Family history, *n* (% yes)				
Myocardial infarction	1383 (57)	647 (53)	736 (61)	*<0.01*
Diabetes	829 (34)	441 (36)	388 (32)	*0.03*
Stroke	986 (41)	473 (39)	513 (42)	*0.11*
Medication use, *n* (%) ^d^	364 (15)	123 (10)	241 (20)	*<0.01*
Hormone usage, *n* (%)				*<0.01*
Current Estrogen + Progesterone	355 (15)	205 (17)	150 (12)	
Current Estrogen alone	555 (23)	295 (24)	260 (21)	
Past Users	375 (15)	175 (14)	200 (17)	
Never Used	1143 (47)	539 (44)	604 (50)	
Hypertension, *n* (%)				*<0.01*
Never hypertensive	1409 (58)	824 (68)	585 (48)	
Untreated hypertensive	237 (10)	96 (8)	141 (12)	
Treated hypertensive	782 (32)	294 (24)	488 (40)	
Diabetes, *n* (% yes)	197 (8)	46 (4)	151 (12)	*<0.01*
Hypercholesterolemia, *n* (% yes)	380 (16)	176 (14)	204 (17)	*0.14*
Hysterectomy, *n* (% yes)	1000 (41)	500 (40)	500 (40)	*Matched*
**Dietary factors**				
Alcohol, g/day ^c^	0.9 (6.5)	1.0 (7.0)	0.6 (6.2)	*0.02*
Percent calories from carbohydrates ^c^	52.2 (13.4)	52.8 (9.4)	51.2 (10.0)	*<0.01*
Percent calories from protein ^c^	16.8 (4.2)	16.9 (4.2)	16.8 (4.2)	*0.63*
Total energy, Kcal/day ^c^	1506.4 (728.3)	1531.3 (702.4)	1482.2 (752.6)	*0.84*
Fish, servings/day ^b^	0.3 (0.2)	0.3 (0.2)	0.2 (0.2)	*0.28*
Dairy products, servings/day ^b^	1.9 (1.5)	1.9 (1.4)	1.9 (1.5)	*0.53*
Butter, teaspoons/day ^b^	0.2 (0.5)	0.2 (0.5)	0.3 (0.6)	*<0.01*
Margarine, teaspoons/day ^b^	0.2 (0.5)	0.2 (0.5)	0.2 (0.5)	*0.41*
Olive/Canola oil, teaspoons/day ^b^	0.1 (0.2)	0.1 (0.2)	0.1 (0.3)	*0.57*
Other vegetable oils, teaspoons/day ^b^	0.1 (0.2)	0.0 (0.1)	0.1 (0.2)	*0.28*
Red meat, servings/day ^b^	0.6 (0.5)	0.6 (0.5)	0.7 (0.6)	*<0.01*
Carbohydrates, g/day ^b^	202.5 (76.1)	204.2 (72.3)	200.8 (79.7)	*0.27*

^a^*p* values were derived using paired *t* test (continuous and normally distributed variables), Wilcoxon signed rank test (continuous non-normally distributed variables), or McNemar test (categorical variables). ^b^ The mean (SD) of normally distributed continuous variables. ^c^ The median (IQR) of continuous non-normally distributed variables. ^d^ Medications included anticoagulant, anti-diabetic, and lipid lowering medications. Abbreviations: BMI (body mass index), CHD (coronary heart disease), IQR (interquartile range), MET-h (metabolic equivalent-hours), SD (standard deviation).

**Table 3 nutrients-11-01672-t003:** Multivariable adjusted associations (1 moL %) between plasma phospholipid fatty acids and CHD risk in the matched case-control study (*N* = 2428).

Plasma Phospholipid Fatty Acids	Mean (SD)	Model 1 ^a^	Model 2 ^b^	
	Mol %	OR (95% CIs)	OR (95% CIs)	OR (99% CIs)
SFA	46.09 (1.29)	1.19 (1.11, 1.28)	1.20 (1.10, 1.30)	1.20 (1.08, 1.34)
Long-chain SFA ^c^	44.74 (1.37)	1.17 (1.09, 1.25)	1.18 (1.09, 1.28)	1.18 (1.07, 1.31)
Very-long-chain SFA ^d^	1.35 (0.46)	1.00 (0.80, 1.26)	1.00 (0.77, 1.30)	1.00 (0.71, 1.41)
MUFA	11.77 (1.64)	0.96 (0.91, 1.01)	0.98 (0.93, 1.04)	0.98 (0.91, 1.06)
PUFA n-3	5.14 (1.58)	0.89 (0.84, 0.94)	0.93 (0.88, 0.99)	0.93 (0.86, 1.01)
PUFA n-6	36.31 (2.46)	1.03 (0.99, 1.06)	1.00 (0.96, 1.03)	1.00 (0.95, 1.05)
TFA	0.69 (0.34)	1.06 (0.84, 1.33)	1.01 (0.78, 1.31)	1.01 (0.72, 1.42)

^a^ Model 1 adjusted for matching factors (age, race/ethnicity, enrollment date, and hysterectomy status). ^b^ Model 2 additionally adjusted for income, lifestyle factors (physical activity and smoking), CHD risk factors (family history of myocardial infarction/diabetes, postmenopausal hormone use, and self-reported hypertension/diabetes), and dietary factors (percent calories from protein/carbohydrates and total energy intake). ^c^ Long-chain SFA included lauric acid (12:0), myristic acid (14:0), pentadecylic acid (15:0), palmitic acid (16:0), and stearic acid (18:0). ^d^ Very-long-chain SFA included arachidic acid (20:0), behenic acid (22:0), and lignoceric acid (24:0). Abbreviations: CHD (coronary heart disease), CIs (confidence intervals), MUFA (mono-unsaturated fatty acids), PUFA (polyunsaturated fatty acids), OR (odds ratio), SD (standard deviation), SFA (saturated fatty acids), TFA (*trans* fatty acids).

**Table 4 nutrients-11-01672-t004:** Odds ratios (95% CIs) of CHD associated with 1 moL % substitutions between plasma phospholipid fatty acid groups among participants in the matched case-control study (*N* = 2428).

Plasma Phospholipid Fatty Acids	Model 1 ^a^	Model 2 ^b^	
	OR (95% CIs)	OR (95% CIs)	OR (99% CIs)
PUFA n-6↓ PUFA n-3 ↑ (1 moL %)	0.85 (0.80, 0.90)	0.90 (0.84, 0.96)	0.90 (0.83, 0.98)
TFA↓ PUFA n-3 ↑ (1 moL %)	0.72 (0.55, 0.94)	0.74 (0.56, 0.99)	0.74 (0.51, 1.09)
TFA↓ PUFA n-6 ↑ (1 moL %)	0.84 (0.64, 1.11)	0.82 (0.61, 1.11)	0.82 (0.56, 1.22)

^a^ Model 1 adjusted for matching factors (age, race/ethnicity, enrollment date, and hysterectomy status). ^b^ Model 2 additionally adjusted for income, lifestyle factors (physical activity and smoking), CHD risk factors (family history of myocardial infarction/diabetes, postmenopausal hormone use, and self-reported hypertension/diabetes), and dietary factors (percent calories from protein/carbohydrates and total energy intake). Abbreviations: CHD (coronary heart disease), CIs (confidence intervals), PUFA (polyunsaturated fatty acids), OR (odds ratio), TFA (*trans* fatty acids).

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
