# Peer review of "Plasma Phospholipid Fatty Acids and Coronary Heart Disease Risk: A Matched Case-Control Study within the Women’s Health Initiative Observational Study"

_nutrients, 2019, doi:10.3390/nu11071672_

Round 1
Reviewer 1 Report
Thank you to the authors for the opportunity to review their work. I have no major concerns about the work; so, I will present only a few minor and moderate points for the authors and editors to consider.
Minor:
Edits: Line 92 - is 'on' the intended word?. Line 108 - is 0.40 an OR? Line 168 - consider moving the (content) to the table description or footnote. Table 1 - consider adding group total percents. Line 208 - WST 'test'. Line 211 - 'hypothesize' causal diagram. Line 224 - the '~' could be interpreted as beta 1 regressed onto beta 4 (maybe use 'to' instead of the '~'). Line 262 - is the OR=1.08 [1.02-1.15] in Table 3? Line 313 - the TFA to PUFA n-6 direction was not the same (1.10 > 1). Line 371 - consider a ';' instead of a ','. Appendix 1 - did you explain the cross after the P in the heading? Line 456 - Carb instead of Carbs? Line 483 - spacing. Line 499 - is 'a' in the actual table? Line 224 - the stated model isn't in the conditional logistic model form.
Is there literature to support your dietary grouping factors (Line 186 to 190)? The discussion adds some, but you may want your choices better defended here too.
Appendix 3 could be improved. Consider bolding significant results. Consider adding to the table and not jus the footnote the type of variable (e.g. matching or demographic). Consider moving the crude and adjusted models to the top or some other way of highlighting them. In the text (~line 213) make it more clear these results are for models with only one PL-FA in the model (is this correct?).
Paragraph 232 to 235. Add the number of complete cases and number missing all family history information.
While maybe not important, the word 'risk' is used to interpret ORs in many places. Positive and negative (lines 261 and 263) could be interpreted as the opposite of what you mean. Some might say positive is a good things so the effect must be to lower odds of CHD.
Line 266 - If the results of individual PF-FAs are elsewhere, should Appendix 3 be compared to these to offer validation of your data?
In Appendix 6 how can both Strat be about .95 but the combined is .74? Please check this result.
Please confirm all needed model assumptions were met.
Realizing they are appendices, the tables are still overwhelming and slightly hard to read (in the sense of knowing exactly which results goes with which column and row). This is especially true when they span multiple pages.
Moderate:
One goal is to find biomarkers. I am used to a more stringent significance level than 5% when claiming association with a biomarker.
I am not convinced of the completeness of the sensitivity analysis for a few reasons: (1) I would want to see different significance levels (different CIs in the manuscript) as per the reason stated above, (2) I am most concerned about sensitivity to the imputations, (3) nonparametric tests are used throughout for single covariate analysis suggesting non-normality or the presence of outliers, but this is not considered in the manuscript's main model, the conditional logistic regression. Influential observations or outliers should have been examined, (4) The different measure of the PL-FA (the main variable of focus) did change the significance of the result forTFA to PUFA n-3 (Appendix 8).
The spearman correlation analysis could be described better. In fairness, its intent became much more clear in the discussion and in rereading the intro (e.g. in line 124) the research goal is stated; however, it is not as clear in the Statistical Analysis section. For example, why the combined versus controls and not cases versus controls and then a test of differences in correlations? In the results did you only report results significant at a 0.05 level (maybe add an asterisk or bold the dietary group with significant correlation in Appendix 4?). More, I interpreted the introduction to mean that one goal was to show that dietary surveys are not a good enough tool; however, the correlation analysis depends on them? Consider justifying the use of dietary surveys more (the description of the impact of their accuracy in lines 370 to 375 was nice though).
I am not convinced the results are novel enough for publication. I appreciate the cohort, matched design, and prospective additions. However, I interpreted one of the authors' main goals was to be one of the first groups to simultaneously look at the different types of PL-FAs as individual PL-FA results had been done already (line 266 and reference 13). However, the authors themselves state (line 398) they only had a subset of FAs that you would want to consider.
Related to 4 above: I think the authors may have modeling issues. If they included all 5 categories of FAs totaling to 100% in the model, then they may have collinearity issues and incorrect coefficient interpretations. PUFA n-3 results in Table 4 seems to me more accurate than Table 3 for this reason, but the main interpretations in the article use Table 3. For example, the ORs for PUFA n-6 and TFA in model 3 in Table 3 or only 0.01 different, but the comparisons to PUFA n-3 in Table 4 are 0.17 different? I may be misinterpreting the article and models. If so, please clarify your final model in the article.
There are so many (initially justified) covariates in the models, which I understand is necessary for a first model, especially in medical studies. However, there is no parsimonious model given. That is, they have potentially included a lot of covariates that aren't associated with CHD (through the model at least) affecting their estimates of the effects of the PL-FAs. Consider adding an appendix giving the full model output if you don't want to find a parsimonious model.
Author Response
Please fine detailed response in the uploaded word file.

Reviewer 2 Report
Qing et al checked fatty acids profile in plasma samples of women having coronary heart disease. PL-FA was measured using gas chromatography. Author proved his hypothesis that there is a link between PL-FA levels with CHD in postmenopausal women. Although this is not a novel study regarding link between fatty acids and heart disease but author did study in menopausal women which makes it different. All experiments are well designed and results are very clear.
Thanks
Author Response
Response: We appreciate the reviewer’s comments and have made the following improvements:
1) We have revised the introduction section to highlight the importance of this paper is to provide more evidence related to fatty acids and CHD from postmenopausal women, which can fill the research gap. We also clarified the advantage of using PL-FA substitution model over dietary substitution in the introduction section. Relevant references have been added into the manuscript.
2) We have proofread the entire manuscript and addressed any language issue once observed.
Reviewer 3 Report
Dear Authors,
Congratulation for this article that will be of great interest, especially for nutritionist. It is clear, and the interpretations are realistic, honest and useful. On our opinion it can be publisher as it is! Only few observations, that may be will be of interest for you to refer in this article. Olive and canola together as food group, does it make sense, one is rich in MUFA the other in n3, may be this association can confound the results? The correlations with carbohydrates consumption should be of interest, because cereals consumptions have been associated with increase in n6 consumption. Also the difference between control and cases were significant for carbohydrates, with equivalent energy intakes the cases group eat less CHO than the controls, does it mean that the exogenous participation of CHO in long chain SFA did not had influences? Other factors of influence like chronic weight cycling, that are not controlled in this study can contribute to the limits of the study.
Author Response
Response: With respect to the reviewer’s concern about olive and canola together as a food group, unfortunately, these “well-known best two” oils were measured together in the food frequency questionnaire of the Women’s Health Initiative study. Therefore, we could not get the accurate breakdowns of these oils, and the correlation we observed might not be accurate since one is rich in MUFA and another in n-3. However, considering the dietary PLFA n-3 is the predominant source of PL-PUFA n-3, the correlation between olive/canola oils and PL-PUFA n-3 is more likely to be underestimated, which means our result is conservative. We have examined the association between dietary carbohydrates and PL-FA, however, no significant association was observed (although we also observed positive correlation between dietary carbohydrates and PUFA n-6 consumption). To account for chronic weight cycling, we performed sensitivity analysis by additionally adjust for 3-year BMI change and our results did not change much (Appendix 5).

Round 2
Reviewer 1 Report
The authors addressed all my concerns and make clarifying edits as needed. There are still a few grammatical errors (e.g. line 117 'were' should be 'was'), but the typesetter will likely correct these as needed and don't warrant requesting another revision.